# Prediction of Chronic Lower Back Pain Using the Hierarchical Neural Network: Comparison with Logistic Regression—A Pilot Study

**DOI:** 10.3390/medicina55060259

**Published:** 2019-06-09

**Authors:** Yutaka Owari, Nobuyuki Miyatake

**Affiliations:** 1Shikoku Medical College, Utazu, Kagawa 769-0205, Japan; 2Department of Hygiene, Faculty of Medicine, Kagawa University, Miki, Kagawa 761-0793, Japan; miyarin@med.kagawa-u.ac.jp

**Keywords:** chronic lower back pain, hierarchical neural network, logistic regression analysis

## Abstract

*Background:* Many studies have reported on the causes of chronic lower back pain (CLBP). The aim of this study is to identify if the hierarchical neural network (HNN) is superior to a conventional statistical model for CLBP prediction. Linear models, which included multiple regression analysis, were executed for the analysis of the survey data because of the ease of interpretation. The problem with such linear models was that we could not fully consider the influence of interactions caused by a combination of nonlinear relationships and independent variables. *Materials and Methods:* The subjects in our study were 96 people (30 men aged 72.3 ± 5.6 years and 66 women aged 71.9 ± 5.4 years) who participated at a college health club from 20 July 2016 to 20 March 2017. The HNN and the logistic regression analysis (LR) were used for the prediction of CLBP and the accuracy of each analysis was compared and examined by using our previously reported data. The LR verified the fit using the Hosmer–Lemeshow test. The efficiencies of the two models were compared using receiver performance analysis (ROC), the root mean square error (RMSE), and the deviance (−2 log likelihood). *Results:* The area under the ROC curve, the RMSE, and the −2 log likelihood for the LR were 0.7163, 0.2581, and 105.065, respectively. The area under the ROC curve, the RMSE, and the log likelihood for the HNN were 0.7650, 0.2483, and 102.787, respectively (the correct answer rates were HNN = 73.3% and LR = 70.8%). *Conclusions:* On the basis of the ROC curve, the RMSE, and the −2 log likelihood, the performance of the HNN for the prediction probability of CLBP is equal to or higher than the LR. In the future, the HNN may be useful as an index to judge the risk of CLBP for individual patients.

## 1. Introduction

The prevalence of lower back pain (LBP) in an adult is approximately 12% [1]. In Japan, in 2013, the proportion of LBP patients was reported as 9.2% (male) and 11.8% (female) [2]. In the Brazilian older population, the prevalence of chronic LBP (CLBP) was 25.4% [3]. CLBP not only brings about a surge in medical expenses, but also results in enormous economic loss in each country because patients are prevented from participating in society. Therefore, researching the cause of CLBP is an urgent task.

Many researchers have reported on the causes of CLBP. For this study, the analysis of survey data was performed using linear models, which included multiple regression analysis, because of the ease of interpretation. The problem with such linear models was that we could not sufficiently consider the influence of nonlinear relations and interactions caused by combinations between independent variables. Needless to say, we introduced previously used nonlinear regression models such as variable transformation, polynomials, and so on. However, during actual analysis, it was difficult to sufficiently assume such requirements in advance.

In this study, we verified that nonlinear coupling using the hierarchical neural network (HNN) [4,5] is superior for predicting the possibility of CLBP as compared with predictions by a conventional statistical model.

## 2. Materials and Methods

### 2.1. Study Design

We performed a neural network simulation study and adopted the following hypothesis: The HNN is superior for predicting the possibility of CLBP as compared with predictions using a conventional statistical model. The subjects for our study were 96 people (30 men aged 72.3 ± 5.6 years and 66 women aged 71.9 ± 5.4 years) who were members at a college health club, as well as prior subjects from a previous investigation [6]. We excluded subjects with specific lower back pain.

Ethical approval for the study was obtained from the Shikoku Medical College Ethics Screening Committee, Japan (approval number: H28-8, 25 May 2016). The decision date of the ethics committee was 25 May 2016.

### 2.2. Clinical Parameters and Measurements

The data included age, gender, sleeping time, spouse, steps, BMI, and excluding ≤1.5 metabolic equivalents (METs). We collected for 3 consecutive days during each week over the study period. Stiff shoulder was defined as pain in the neck, shoulder, or back, and chronic stiff shoulder was pain lasting longer than 3 months. In addition, chronic stiff shoulder was considered as a variable affecting psychological distress. We used a visual analog scale as a measurement tool to investigate the degree of pain experienced by each subject. The scale consisted of a straight line where the degree of pain was expressed (scored) as a numerical value from 0 to 10 points. The left end of the straight line was “painless”, and the right end was “the most severe pain that can be imagined”. Subjects were asked to place a cross on the line at a point that seemed to be the degree of pain they were feeling. Assuming that the total length of the straight line was 10, the distance from the left end to the cross indicated the degree of pain. The closer the number was to 10, the stronger the pain. The subjects were requested to respond by choosing either “experienced” (1, other than 0), or “not experienced” (0) [6]. 

### 2.3. Psychological Distress

For the data from a previous investigation, we used the K6 scale. The K6 is a self-written questionnaire developed by Kessler as a screening test for psychological distress that could effectively discriminate psychological distress [7]. It is valid and reliable. Subjects answered six items on a 5-point Likert scale, and responses for each item were transformed to scores ranging from 0 to 4 points. The questionnaire consisted of the following six questions: Over the last month, about how often did you feel: (1) nervous, (2) hopeless, (3) restless or fidgety, (4) so sad that nothing could cheer you up, (5) that everything was an effort, or (6) worthless? For each of the questions, the subjects were requested to choose from the following responses: All of the time (4 points), most of the time (3 points), some of the time (2 points), a little of the time (1 point), and none of the time (0 point). The total of the responses was the evaluation level [8]. Thus, the score range was 0–24. A higher total score corresponded to higher psychological distress. 

### 2.4. Chronic Lower Back Pain (CLBP)

Chronic lower back pain was defined as pain localized between the 12th rib and the inferior gluteal folds [9] lasting longer than 3 months [10]. The subjects were shown the location of the pain and requested to respond to chronic lower back pain entered on the questionnaire by choosing from the following: “experienced” (1), “not experienced” (0) [6]. In addition, the subjects received a definitive diagnosis of orthopedic surgery.

### 2.5. Social Participation

Using the data from a previous investigation, we evaluated social participation as established by Haeuchi et al. [11]. The respondents were asked whether they participated in eight types of social activities. These types were as follows: (1) local events and festivals; (2) resident and neighborhood associations; (3) circle activities, i.e., a group activity based on a hobby or an interest such as history; (4) golden age club; (5) volunteer activities; (6) religious activities; (7) paid work; and (8) learning in a social environment within the past year from the date of the survey. The subjects were requested to respond by choosing either “some kind of social activity or more than one within the past month, once or more per week” (1) or “participates in no activities” (0) [6].

### 2.6. Models

#### 2.6.1. HNN

Variables: For this study, there were two dependent variables for CLBP, i.e., absence (0, experienced) and presence (1, not experienced). In order to carry out the comparison with the logistic regression analysis, the number of independent variables was set to four. There were 10 possible candidates for the independent variables which included gender [12], age [13], spouse (presence or none), BMI (kg/m^2^) [14], chronic stiff shoulder (presence or none), social participation (presence or none) [15,16,17], sleeping time (hours/day), the K6 scores [18,19,20], a rate of 1.5 METs or less [21], and number of steps (steps/day). The relationship among these ten independent variables, in a conventional study, and the dependent variables was discussed, and the four variables with high correlation were taken as the independent variables. Therefore, the four independent variables selected were age, body mass index (BMI), social participation and the K6 score.

Structure of HNN: The neural network consisted of multiple layers (input layer, hidden layer, and output layer). The number of nodes of the input layer was the number of nodes of data 4. Moreover, by changing the number of layers in the hidden layer and the number of nodes in each layer, we identified how the accuracy changed. We adopted the number of layers of the hidden layer and the number of the nodes of each layer which were higher in the system. Here, we adopted a sigmoid function as the activation function of the hidden layer. This was the function to output by deforming the input data (transform). For the prediction of CLBP, we determined the number of output layers to be two (2 nodes, onset as 1 = with CLBP and non-onset as 0 = without CLBP). The learning of the neural network was conducted 10,000 times using the back-propagation method.

#### 2.6.2. Logistic Regression Analysis (LR)

We used LR to compare with HNN, using the four independent variables discussed above.

### 2.7. Statistical Analyses

First, we used the HNN to determine the number of hidden layers and the number of nodes in each layer by trial and error. Second, the HNN was used to verify the prediction rate of the onset of CLBP. Third, the efficiencies of the two models were compared using the receiver performance analysis (ROC), the root mean square error (RMSE), and the deviance (−2 log likelihood). In addition, we compared the prediction accuracy rates of the two models. And finally, we increased the independent variables to 10 by adding gender, spouse (presence or absence), shoulder stiffness, sleeping time, rate of 1.5 METs or less, and steps. We verified the prediction rate of CLBP onset with the HNN using the software SPSS ver. 24.0 (IBM SPSS Inc., Chicago, IL, USA) and Python 3.6.4 (Python Software Foundation, Delaware, United States).

## 3. Results

The subject characteristics are summarized in Table 1. First, the model had the highest accuracy when the number of hierarchical layers was one and the number of nodes was seven (Figure 1). Second, Table 2 compares the HNN and the LR. Using the case for learning, of the 33 with CLBP, 18 were accurately predicted, and of the 33 without CLBP, 31 were accurately predicted. Of the total 66, 49 were accurately predicted. In learning, the result was accurately predicted overall (74.2%). When the learning model was applied to the testing model, the accuracy answer rate was 73.3% and the accuracy rates obtained for the HNN and the LR were 73.3% and 70.8%, respectively. Third, the area under the ROC curve, the RMSE, and the −2 log likelihood for the LR were 0.7163, 0.2581, and 105.065, respectively, and for HNN they were 0.7650, 0.2483, and 102.787, respectively (Table 3 shows the area under the ROC between HNN and LR in detail). And finally, the predictive value rate increased from 73.3% to 84.6% using 10 variables (Table 4).

## 4. Discussion

The HNN showed prediction ability equal to or greater than the LR. In the future, the HNN is expected to be useful as an index to judge the risk of CLBP for individual patients. First, in the input layer, when the order of the inputting variables was changed, the initial values of various patterns were assigned, and therefore the result might be affected. Therefore, the order of various variables was changed, and the stability of a specific solution was evaluated. We found the most accurate model by changing the total number of hidden layers. As a result, when looking at the accuracy with the verification data, the model with one layer of a hidden layer has the highest accuracy. Moreover, we examined the number of nodes with the highest accuracy with the model of the selected hidden layer being one layer. In this case, if the number of nodes in the hidden layer increased, the accuracy with respect to the learning data increased, but the accuracy of estimation with the testing data was not necessarily high. As a result, seven nodes had the highest accuracy (Figure 1). Secondly, the HNN did not require specifying a prior model and could predict from patterns learned in the data [21]. In addition, even if there was a process with a large unexpected influence in advance, the influence could be effectively found and detected [21,22]. Therefore, it was possible to predict more accurately than the LR model. Finally, the independent variables were increased to 10 by adding gender, spouse (presence or absence), shoulder stiffness, sleeping time, rate of 1.5 METs or less, and steps. The predictive value rate increased from 73.3% to 84.6%. Improvement of the prediction rate also depended on the height of the correlation coefficient between the dependent and independent variables.

The potential limitations of this study are as follows: First, the responses reported by the individual participants with respect to their response of chronic lower back pain were not always accurate, and therefore an orthopedic surgeon was contacted to act as an adviser for the health club. Although factors such as work history, education, income, etc., are considered to be important causes of chronic lower back pain, they could not be considered. Second, due to the small sample size, verification of this survey was somewhat difficult. When the number of samples is small, the results obtained become unstable. For this study, the number of samples was 96, which is considered small, and therefore the following countermeasures were taken: Samples were randomly classified into model learning and verification and random numbers were changed and performed 10,000 times. The average value of each sample, which was plurality estimated, may not represent an appropriate estimated value. Future investigations will need to increase the number of samples and conduct rigorous verification. Third, although the HNN has high prediction ability, it cannot explain the mutual relationship between dependent and independent variables [23]. Empirical research is necessary to explain the mutual relationship.

## 5. Conclusions

On the basis of the ROC curve, the RMSE, and the −2 log likelihood, the prediction probability performance of the HNN for predicting CLBP is equal to or higher than the LR. In the future, the HNN is expected to be useful as an index to judge the risk of CLBP for individual patients.

## Figures and Tables

**Figure 1 medicina-55-00259-f001:**
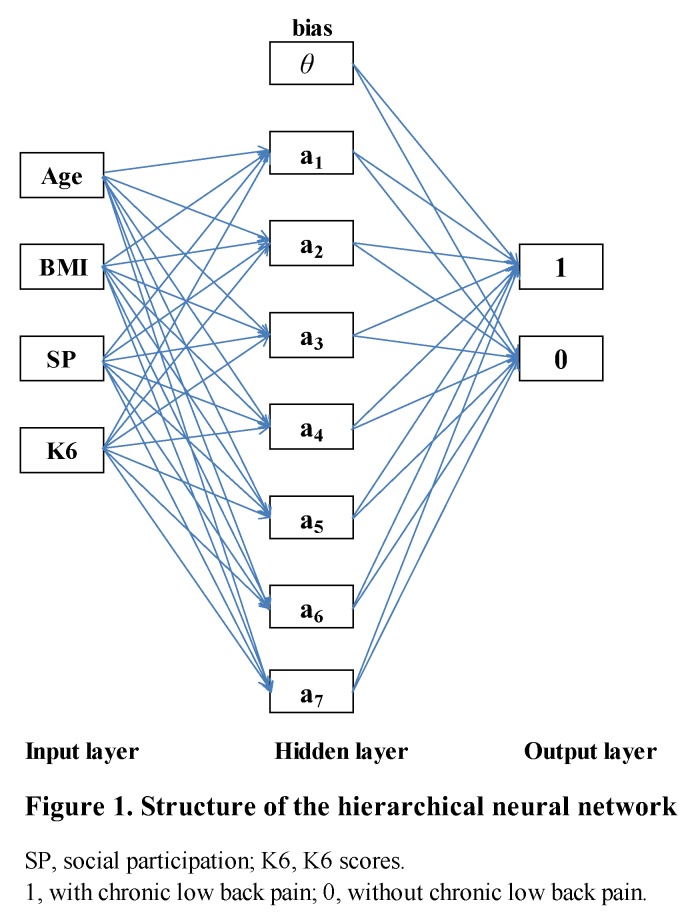
The rectangles indicate nodes, and weights are indicated by arrows (→) between nodes. The number of nodes in the input layer is 4 (age, BMI, social participation (SP), and K6 scores), the number of nodes in the hidden layer is 7 (7 hidden neurons), and the number of nodes in the output layer is 2 (1: With chronic lower back pain, 0: Without chronic lower back pain).

**Table 1 medicina-55-00259-t001:** Clinical characteristics of the enrolled subjects in a college health club in the Kagawa Prefecture (Japan).

	Total					Men					Women				
	Mean	±	SD	Min	Max	Mean	±	SD	Min	Max	Mean	±	SD	Min	Max
Number of Subjects	96					30					66				
Age (year)	72.0	±	5.4	65	85	72.3	±	5.6	65	85	71.9	±	5.4	65	85
BMI (kg/m^2^)	22.7	±	2.9	14.9	30.2	23.4	±	2.9	17.6	29.1	22.4	±	2.8	14.9	30.2
Number of Steps (steps/day)	5692.8	±	2527.2	569.9	12230.1	5881.5	±	2413.8	1585.4	11049.1	5615.5	±	2587.7	569.9	12230.1
Sleep Time (hours/day)	6.5	±	1.1	4	10	6.7	±	0.9	5	9	6.5	±	1.1	4	10
≤1.5 METs (%/day)	55.9	±	10.0	35.4	79.9	59.5	±	11.6	36.8	79.9	54.3	±	8.9	35.4	75.7
K6 Scores	2.7	±	3.3	0	14	3.1	±	3.6	0	13	2.5	±	3.2	0	14
Spouse (Present) (%)	75.5					96.0					67.2				
Social Participation (Participates) (%)	83.3					76.7					86.4				
Chronic Stiff Shoulder (Present) (%)	26.0					16.7					30.3				
Chronic Low Pack Pain (Present) (%)	47.9					50.0					47.0				

Min: Minimum; Max: Maximum; BMI: Body mass index (kg/m^2^), METs: Metabolic equivalents.

**Table 2 medicina-55-00259-t002:** Comparison between the hierarchical neural network (HNN) and logistical regression (LR). Classification of the results (HNN): 66 learning data, 49 (18 + 31) accurately predicted, and 30 testing data, 22 (14 + 8) accurately predicted.

	**Actual**	**Predicted**		
		**Non-Default**	**Default**	**Accuracy (%)**
Learning	Non-Default	31	2	93.9
	Default	15	18	54.5
	Total			74.2
Testing	Non-Default	14	3	82.4
	Default	5	8	61.5
	Total			73.3
LR				
	**Actual**	**Predicted**		
		**Non-Default**	**Default**	**Accuracy (%)**
	Non-Default	38	12	76.0
	Default	16	30	65.2
	Total			70.8

HNN: Hierarchical neural network; LR: Logistic regression; Non-Default: Without CLBP, Default: with CLBP; HNN: *N* = 30; testing; LR: *N* = 96.

**Table 3 medicina-55-00259-t003:** Comparison between HNN and LR by the area under the ROC curve.

	Area	Standard error	Lower Bound	Upper Bound	*p*
HNN	0.7650	0.0492	0.6552	0.8315	<0.001
LR	0.7163	0.0557	0.6070	0.8255	<0.001

HNN: Hierarchical neural network; LR: Logistic regression.

**Table 4 medicina-55-00259-t004:** Relationship between each independent variable.

	Gender	Age	Spouse	BMI	CSS	SP	Sleep Time	K6 Scores	1.5METs or Less	Steps
Gender	1.0000	0.2000	0.2957	0.0949	0.1426	0.2372	0.0980	0.0458	0.2896	0.0265
Age		1.0000	0.0728	0.0344	0.1725	0.0959	−0.0956	0.0392	0.2285	−0.3479
Spouse			1.0000	0.0509	0.0426	0.1176	0.2946	0.1091	0.0547	0.0624
BMI				1.0000	0.0994	0.0224	0.0509	−0.0334	0.2848	−0.1740
CSS					1.0000	0.2372	0.0100	0.0721	0.1077	0.1889
SP						1.0000	0.0883	0.1034	0.0728	0.0616
Sleep Time							1.0000	0.0025	−0.2184	0.0054
K6 Scores								1.0000	0.1020	−0.2173
1.5METs or Less									1.0000	−0.2956
Steps										1.0000

CSS: Chronic stiff shoulder; SP: Social participation; Correlation coefficient: Quantitative variable vs. quantitative variable; Correlation ratio (η): Quantitative variable vs. qualitive variable; Coefficient of association: Qualitative variable vs. qualitative variable.

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
