# Peer review of "Prediction of Chronic Lower Back Pain Using the Hierarchical Neural Network: Comparison with Logistic Regression—A Pilot Study"

_medicina, 2019, doi:10.3390/medicina55060259_

Round 1
Reviewer 1 Report
This paper is very fluid and explains it's objectives clearly. I think the literature is much richer than referenced although their chosen three models HNN, LR, LDA may be unique. I have not examined that.
The restricted age group is of my concern and the small sample size in this study. I am not convinced of adequacy of sample size, population sample appropriate for low back pain, in light of natural history of low back pain.
Some references such Bigos et al. and Marras et al., may be illuminating.
Good luck to author.
Author Response
Comment 1
This paper is very fluid and explains it's objectives clearly. I think the literature is much richer than referenced although their chosen three models HNN, LR, LDA may be unique. I have not examined that.
(Answer) There are many models, but few models can be compared directly with HNN. LDR is also excluded from the title because it cannot be compared directly with HNN. Also, we added the main assumptions to abstract, statistical analyses, and result as follows.
(Old Title)
Prediction of Chronic Low Back Pain Using Hierarchical Neural Network: Comparison with Logistic Regression and Linear Discriminant Analysis Model
(New Title)
Prediction of Chronic Low Back Pain Using Hierarchical Neural Network: Comparison with Logistic Regression - A pilot study -
Lines 19-24.
Abstract
LR verified the fit with Hosmer and Lemeshow test. The efficiencies of the two models were compared by receiver performance analysis (ROC), root mean square error (RMSE) and Deviance (-2log-likelihood). Results: The area under ROC curve, RMSE and -2log-likelihood of LR was 0.7163, 0.2581 and 105.065, respectively. The area under the ROC curve, RMSE and -2log-likelihood of HNN was 0.7650, 0.2483 and 102.787, respectively (the correct answer rate: HNN = 73.3%, LR = 70.8 %).
The restricted age group is of my concern and the small sample size in this study. I am not convinced of adequacy of sample size, population sample appropriate for low back pain, in light of natural history of low back pain.
(Answer) Back pain increases with age. it is 25.5% in 20's, but it increases to 35.8% in 70's in Japan (National research report on low back pain: Japan Orthopedic Association). I think that I investigated and analyzed in the future. And, the small number of samples is a drawback of this study. Therefore, we added the following supplement to the discussion part.
Lines 170-176.
The potential limitations of this study are as follows. First, due to the small sample size, verification of this survey is somewhat difficult. If the number of samples is small, the result obtained becomes unstable. This time, the number of samples was as small as 96, so the following countermeasures were taken. That is, samples were randomly classified into model learning and verification, and random numbers were changed and performed 10,000 times. The average value of each sample, which is plurality estimated, might not be an appropriate representative estimated value. It will be necessary to increase the number of samples and conduct rigorous verification.
Some references such Bigos et al. and Marras et al., may be illuminating.
(Answer) Linear and Poisson regression models (Marras et al.) are useful for risk assessment. However, we do not consider these models this time, because they cannot be compared directly with HNN. I could not refer to Bigos et al.
Good luck to author.

Reviewer 2 Report
The work requires thorough changes.
1. The main assumption of research and study design is not clearly formulated.
In this study, authors have verified whether nonlinear coupling by hierarchical neural network (HNN) is superior to prediction by conventional statistical model when predicting the possibility of CLBP, but this is not a clearly defined goal that will be clear and understandable to the reader.2. Methods. What scale of pain assessment did the authors use in their study? Are questionnaires validated, which one was used?3. Results. There is no statistical analysis of the results obtained. The authors examined a lot of different factors characterizing the studied population, but they did not present whether they have statistical significance for the presented hypothesis. Presentation of the results on graph 1 is unclear and needs improvement. The results in tables 1, 2 and 3 also need to be specified and commented.4. Discussion The authors did not present the results in principle. The reference to three other researchers' work is not enough. The discussion of results is a total change. One should look for a literature addressing the discussed topic. Presentation of the use of a hierarchical neural network in other medical research will support the research proposed by the authors.5. The authors presented the potential limitations of this study, although a small number of respondents is not the greatest limitation of the presented study. On the other hand, the authors did not present a single strength of the obtained results.6. The work requires thorough changes.Author Response
Comment 2
1. The main assumption of research and study design is not clearly formulated.
In this study, authors have verified whether nonlinear coupling by hierarchical neural network (HNN) is superior to prediction by conventional statistical model when predicting the possibility of CLBP, but this is not a clearly defined goal that will be clear and understandable to the reader.
(Answer) LDR is also excluded from the title because it cannot be compared directly with HNN. Also, we added the main assumptions to abstract and statistical analyses as follows.
(Old Title)
Prediction of Chronic Low Back Pain Using Hierarchical Neural Network: Comparison with Logistic Regression and Linear Discriminant Analysis Model
(New Title)
Prediction of Chronic Low Back Pain Using Hierarchical Neural Network: Comparison with Logistic Regression - A pilot study -
Lines 19-24.
Abstract
LR verified the fit with Hosmer and Lemeshow test. The efficiencies of the two models were compared by receiver performance analysis (ROC), root mean square error (RMSE) and Deviance (-2log-likelihood). Results: The area under ROC curve, RMSE and -2log-likelihood of LR was 0.7163, 0.2581 and 105.065, respectively. The area under the ROC curve, RMSE and -2log-likelihood of HNN was 0.7650, 0.2483 and 102.787, respectively (the correct answer rate: HNN = 73.3%, LR = 70.8 %).
3. Statistical Analyses
Lines 116-118.
Also, the efficiency of two models was compared by receiver performance analysis (ROC), root mean square error (RMSE) and Deviance (-2log-likelihood).
2. Methods. What scale of pain assessment did the authors use in their study? Are questionnaires validated, which one was used?
(Answer) We added chronic low back pain as follows.
Lines 50-51.
We excluded subjects with specific low back pain.
Lines 77-78.
The subjects were illustrated the location of the pain and requested to respond to chronic low back pain entered on the questionnaire by choosing from the following: “experienced” (1), “not experienced” (0) [6].
Lines 168-172.
First, as for chronic low back pain, I could not get enough accuracy by the report of the participant alone, so I got an appointment with an orthopedic surgeon who is an adviser of the health class. However, factors such as work history, education, income, etc. are considered to be important causes of chronic low back pain, but this could not be considered.
3. Results. There is no statistical analysis of the results obtained. The authors examined a lot of different factors characterizing the studied population, but they did not present whether they have statistical significance for the presented hypothesis. Presentation of the results on graph 1 is unclear and needs improvement. The results in tables 1, 2 and 3 also need to be specified and commented.
(Answer) We added as follows.
4. Results
Lines 129-134.
Also, the area under ROC curve, RMSE and -2log-likelihood of LR was 0.7163, 0.2581 and 105.065, respectively. The area under the ROC curve, RMSE and -2log-likelihood of HNN was 0.7650, 0.2483 and 102.787, respectively. In Table 3-1, we showed comparison between HNN and LR with the test data. The HNN model was superior to logistic regression model. In Table 3-2, the correct answer rate: HNN = 73.3%, LR = 70.8 %.
Figure 1.
Figure 1. The rectangles indicated nodes, and weights were indicated by arrows (→) between nodes. The number of nodes in input layer was 4 (Age, BMI, SP, and K6), the number of nodes in hidden layer was 7 (7 hidden neurons), and the number of nodes in output layer was 2 (1; with chronic low back pain, 0; without chronic low back pain).
Table 1.
Table 1. Clinical characteristics of the enrolled subjects in a college health club in Kagawa Prefecture (Japan).
Table 2.
Table 2. Classification of the results (hierarchical neural network): .learning data: 66 data, 49 data (18 + 31) was correctly predicted, testing data: 30 data, 22 data (14 + 8) was correctly predicted.
Table 3-1, 3-2.
4. Discussion The authors did not present the results in principle. The reference to three other researchers' work is not enough. The discussion of results is a total change. One should look for a literature addressing the discussed topic. Presentation of the use of a hierarchical neural network in other medical research will support the research proposed by the authors.
(Answer) As mentioned above, along with rewriting the results, we also changed references.
Reference 22, 23.
Mehdi, S.; Hashemi, S.; Kazemnejad, A.; Email author, Lucas, C.; Badie, K. Predicting the type of pregnancy using artificial neural networks and multinomial logistic regression: a comparison study. Neural Computing & Applications, 2005, 14, 198-202.
Eftekhar, B.; Mohammad, K.; Eftekhar, H.A.; Ghodsi, M.; Ketabchi, E. Comparison of artificial neural network and logistic regression models for prediction of mortality in head trauma based on initial clinical data. BMC Med Inform Decis Mak. 2005, S: 3.
5. The authors presented the potential limitations of this study, although a small number of respondents is not the greatest limitation of the presented study. On the other hand, the authors did not present a single strength of the obtained results.
(Answer) We added the main assumptions to abstract, statistical analyses, and results as above.
6. The work requires thorough changes.
(Answer) We have rewritten our manuscript.

Round 2
Reviewer 2 Report
The authors made corrections that significantly increased the value of the work. It was particularly important to describe the detailed scale of the pain assessment used by the authors at work.It is also beneficial to edit tables 2 and 3. The work is interesting, the authors described in more detail the limitations at work.
There is still no indication of the strengths of the study performed. Conclusion should be more detailed, rather than consist of 1 sentence.
Author Response
Thank you for your email. Comments in the second round from the reviewer
are:
---------------------------------------------------------
"The authors made corrections that significantly increased the value of the work. It was particularly important to describe the detailed scale of the pain assessment used by the authors at work.
It is also beneficial to edit tables 2 and 3. The work is interesting, the authors described in more detail the limitations at work.
There is still no indication of the strengths of the study performed. Conclusion should be more detailed, rather than consist of 1 sentence."
(Answer) We added the follows:
Lines: Conclusion; 24-27 (Abstract )and also 185-187 (Conclusion).
Based on ROC curve, RMSE, -2Loglihood, and a prediction probability, HNN might have for CLBP equal to or higher performance than LR. From now on, HNN might be expected to be useful as an index to judge the risk of CLBP of individual patients.
